# High-Throughput Analysis of Leaf Chlorophyll Content in Aquaponically Grown Lettuce Using Hyperspectral Reflectance and RGB Images

**DOI:** 10.3390/plants13030392

**Published:** 2024-01-29

**Authors:** Mohamed Farag Taha, Hanping Mao, Yafei Wang, Ahmed Islam ElManawy, Gamal Elmasry, Letian Wu, Muhammad Sohail Memon, Ziang Niu, Ting Huang, Zhengjun Qiu

**Affiliations:** 1School of Agricultural Engineering, Jiangsu University, Zhenjiang 212013, China; 1000006483@ujs.edu.cn (M.F.T.); wangyafei918@ujs.edu.cn (Y.W.); msohail@ujs.edu.cn (M.S.M.); 2College of Biosystems Engineering and Food Science, Zhejiang University, Hangzhou 310058, China; niu_ziang@zju.edu.cn (Z.N.); t3huang@zju.edu.cn (T.H.); zjqiu@zju.edu.cn (Z.Q.); 3Department of Soil and Water Sciences, Faculty of Environmental Agricultural Sciences, Arish University, North Sinai 45516, Egypt; 4Agricultural Engineering Department, Faculty of Agriculture, Suez Canal University, Ismailia 41522, Egyptgamal.elmasry@agr.suez.edu.eg (G.E.); 5Institute of Agricultural Mechanization, Xinjiang Academy of Agricultural Sciences, Urumqi 830091, China; 6Department of Farm Power and Machinery, Faculty of Agricultural Engineering, Sindh Agriculture University, Tandojam 70060, Pakistan

**Keywords:** aquaponics, AutoML, chlorophyll, hyperspectral reflectance, vegetation indices

## Abstract

Chlorophyll content reflects plants’ photosynthetic capacity, growth stage, and nitrogen status and is, therefore, of significant importance in precision agriculture. This study aims to develop a spectral and color vegetation indices-based model to estimate the chlorophyll content in aquaponically grown lettuce. A completely open-source automated machine learning (AutoML) framework (EvalML) was employed to develop the prediction models. The performance of AutoML along with four other standard machine learning models (back-propagation neural network (BPNN), partial least squares regression (PLSR), random forest (RF), and support vector machine (SVM) was compared. The most sensitive spectral (SVIs) and color vegetation indices (CVIs) for chlorophyll content were extracted and evaluated as reliable estimators of chlorophyll content. Using an ASD FieldSpec 4 Hi-Res spectroradiometer and a portable red, green, and blue (RGB) camera, 3600 hyperspectral reflectance measurements and 800 RGB images were acquired from lettuce grown across a gradient of nutrient levels. Ground measurements of leaf chlorophyll were acquired using an SPAD-502 m calibrated via laboratory chemical analyses. The results revealed a strong relationship between chlorophyll content and SPAD-502 readings, with an $R^{2}$ of 0.95 and a correlation coefficient (r) of 0.975. The developed AutoML models outperformed all traditional models, yielding the highest values of the coefficient of determination in prediction ($R_{p}^{2}$) for all vegetation indices (VIs). The combination of SVIs and CVIs achieved the best prediction accuracy with the highest $R_{p}^{2}$ values ranging from 0.89 to 0.98, respectively. This study demonstrated the feasibility of spectral and color vegetation indices as estimators of chlorophyll content. Furthermore, the developed AutoML models can be integrated into embedded devices to control nutrient cycles in aquaponics systems.

## 1. Introduction

Aquaponics is a non-soil-based, innovative, intelligent, and sustainable agricultural production system combining aquaculture and hydroponics in one system. In aquaponics, plant nutrients are exclusively derived from fish excrement [1]. Briefly, fish excrete their waste, which is instantly transformed into nutrients by nitrifying bacteria, and plants absorb such nutrients, as shown in Figure 1 [1]. Despite adhering to the established design principles for aquaponics, plants frequently experience nutrient insufficiency. The restricted nutrition in the aquaponics system affects the chlorophyll content of plant leaves and canopies. Chlorophyll estimation provides comprehensive knowledge of plants’ nutrient status and nitrogen content [2]. Chlorophyll is essential for precision agriculture as it is the vital pigment in photosynthesis. Furthermore, it is a good indicator of mutations, stress, and nutritional status [3]. Understanding the chlorophyll content of plants is crucial for guiding plant cultivation management, such as determining the amount and timing of fertilization [4].

Chlorophyll is traditionally measured visually and in the laboratory (oven drying and solvent extraction followed by spectrophotometric determination, respectively). Visual methods of estimating chlorophyll content require much experience and may lead to suffering from misleading results among analysts [5]. The conventional laboratory procedures for determining chlorophyll concentration are exceptionally precise. On the other hand, it precludes tracking the dynamic changes in chlorophyll content in plant leaves during the growing season. Moreover, these methods are destructive, time consuming, expensive, and labor intensive [6,7,8]. Consequently, these methods are inappropriate in many cases such as dynamic detection. The SPAD-502 m (soil plant analysis development) was developed to estimate chlorophyll in a rapid, non-destructive, and real-time manner. In addition, it enables us to track dynamic changes in chlorophyll content in plants [6]. Many scientific contributions have established a strong positive relationship between chlorophyll content in crops and SPAD-502 readings [8,9,10,11,12,13]. Recently, SPAD-502 m has been widely used for the estimation of crop chlorophyll content such as lettuce [14,15], wheat [16,17], maize [18,19], apple [20], sugarcane [21], and rice [22].

In recent years, spectral sensing and machine vision have emerged as viable options for crop management and yield estimation [23,24]. Additionally, their potential to facilitate high-throughput plant phenotyping endeavors has garnered significant interest [25]. Significantly, narrowband hyperspectral assessment holds promise in providing a dependable, expeditious, economically viable, and non-destructive method for evaluating the primary photosynthetic pigments in foliage over a large area [26]. The spectral properties of plants are a reliable indicator of leaf surface properties, internal structure, and biochemical properties. Several VIs have been extracted from spectral data using different mathematical relationships (e.g., simple ratios, differences, standard differences, derivatives) to characterize some vegetation features. Importantly, VIs, especially those extracted from the visible to near-infrared (VNIR) and shortwave infrared (SWIR) regions of the electromagnetic spectrum, have demonstrated their worth in estimating plant biomass, physiological properties, and biochemical components [6,27]. Furthermore, VIs are effective tools for identifying spatial and seasonal variations in green vegetation, rendering them well suited for implementation in precision agriculture and crop phenotyping [6]. 

Profiling high-throughput plant phenotyping (HTPP) based on imaging techniques has become widely applied in the research community. Digital images (such as RGB images) can evaluate the state of chlorophyll content in crops by measuring the intensity of reflection in the red, green, and blue bands [18]. Zhang et al. reported that the three primary colors (red, green, and blue) could be used to rapidly estimate chlorophyll in leaves of regenerated crops [28]. Image-based features contain valuable information about plant morphological and biochemical traits, which helps fill the gap between phenotype and genotype for plant improvement [29]. Some algorithms have been developed to determine the relationship between chlorophyll content and color features. Mahmoodi et al. photographed the leaves of four commercial plants with a digital camera to determine chlorophyll content. The results revealed that Kawashima index (IKAW) ((R − B)/(R + B)) is the most fit RGB model for estimating chlorophyll content [30]. 

Apart from the applications of hyperspectral sensors and machine vision techniques, to enhance the understanding of plant behavior, the tremendous development in big data analysis combined with advances in computational power has opened innovative venues for building new techniques for extracting information from plants [31]. Recently, machine learning approaches have been applied, with the help of spectral and image data, as input to develop estimation models for crop traits [32]. Applying machine learning algorithms offers a quick, non-intrusive, and non-destructive approach to quantifying the biochemical constituents of plants. To build a successful machine learning system, one must possess a thorough grasp of mathematical principles and significant proficiency in choosing model architectures [33]. The proficiency of the system relies on the optimal integration of several components, including feature extraction, feature selection, and regression techniques [33]. Hence, finding the system that exhibits optimal performance necessitates a substantial investment of time in trial-and-error experimentation as well as the expertise of a proficient team to assess and evaluate various configurations and models manually. Furthermore, it is imperative to regularly retrain the prediction model due to the substantial variations that might occur among different crops, biochemical components, and geographies. Hence, automatically generating a context-specific machine learning model, even by non-experts, will be a significant difficulty.

Recently, automated machine learning (AutoML) systems have emerged to address these issues by enabling computers to determine automatically the best-suited machine learning path that matches a given task and dataset [33]. Lately, there has been explosive growth in processing power and the accessibility of cloud computing resources. As a result, AutoML has garnered substantial interest from industry and academia. AutoML is an emerging area of research that aims to automate the development of ready-to-use end-to-end ML models with little to no user ML knowledge [34]. AutoML offers a compelling alternative to manual machine learning approaches by offering the potential to produce efficient and comprehensive machine learning pipelines. These pipelines encompass various stages, including data preparation (such as cleaning and preprocessing), feature engineering (such as extraction, selection, and construction), model generation (including selection and hyperparameter tuning), and model evaluation. Importantly, AutoML aims to minimize the user effort and intervention required throughout these processes [35]. AutoML services have become standard offerings in many technology companies, for example, Cloud ML by Google and SageMaker by Amazon.

The feasibility of AutoML has already been established. However, there remains an unignorable domain gap between different agricultural systems. Specifically, the characteristics of aquaponics plants are entirely distinct from those of other agricultural systems. Therefore, the development of models specifically designed for aquaponics plant datasets is imperative. Hence, this study uses SVIs and CVIs to develop and evaluate an AutoML-based model for chlorophyll quantification in aquaponically grown lettuce. Thereby, the following objectives were guided in the present study. First, the SPAD-502 m was calibrated via laboratory analysis to confirm its reliability. Then, an AutoML model leveraging the capabilities of the completely open-source AutoML framework EvalML was developed. Third, we focused on evaluating the reliability of spectral and CVIs, assessing them both separately and in combination to determine their effectiveness as chlorophyll estimators. Fourth, our research comprehensively evaluated the developed AutoML model’s performance, contrasting it with traditional machine learning models. Lastly, we aimed to identify the most effective combination of vegetation indices to achieve the highest possible accuracy in prediction.

Furthermore, to the best of our knowledge, this paper represents the first study to utilize an AutoML system with multispectral vegetation indices derived from an ASD FieldSpec 4 Hi-Res spectroradiometer, alongside color indices obtained from a digital camera, for the estimation of chlorophyll in aquaponically grown plants. Therefore, this study provides new insights and a pathway towards automating and adopting sustainable aquaponics systems as precision agriculture technology. The AutoML developed model can be integrated into embedded devices to control nutrient cycles in aquaponics.

## 2. Materials and Methods

### 2.1. Design of Aquaponics System

An aquaponics system was designed and constructed according to the standard construction standards set by Somerville, Cohen [36]. Since lettuce (romaine lettuce (var. *longifolia*)) is the primary and most popular crop in aquaponics systems, it was used in this study’s research experiments [37]. The two main units of the aquaponics system are the 1 m^3^ fish tank and the plant growing unit. Common carp fish (*Cyprinus carpio* L.) is one of the most popular species cultured in aquaponics systems and was therefore selected [36]. The optimal water quality parameters for this species were provided in terms of temperature (25–30 °C), total ammonia/nitrogen (<1 mg L^−1^), nitrite (<1 mg L^−1^), and dissolved oxygen (>4 mg L^−1^) [36]. The nutrient film technique (NFT) was used for the plant growing unit. Four polyvinyl chloride (PVC) pipes, each 11 cm in diameter and 4 m long, were used as a planting unit to support the plants. Some holes were drilled in these tubes with dimensions suitable for lettuce, and then plastic net cups were placed in these holes, where the seedlings were transplanted. However, a clarifier was installed in the system to purify the water’s large impurities. Additionally, a biological filter, supported by bio-balls as a medium, was used to stimulate the growth of bacteria. Hence, the water flows into the planting tubes, where the plants absorb and purify nutrients from the water. Finally, the water returns to the aquariums by gravity to begin a new cycle, etc. Notably, the fish tank was covered with a thin plastic net and not exposed to direct sunlight, while the plant unit was fully exposed to direct sunlight [38]. Figure 1 shows the schematic diagram of aquaponics system. All aquaponics parameters were monitored and maintained at optimum levels for system equilibrium. A hydroponic (control) system has been constructed and designed with Hoagland’s full-strength nutrient solution [39]. This system provided all of the optimal conditions for plant growth compared with plants growing in the aquaponics system. 

The proposed methodology followed in this study is illustrated in the diagram shown in Figure 2. Since chlorophyll includes most plant nitrogen, different nitrogen concentrations were used to prepare four different nutrition levels. The control system (A) received a full-strength Hoagland solution. For the second system (B), 600 mg/L of NO_3_-N was added to the aquaponics solution. In the third system (C), 300 mg/L of NO_3_-N was added to the aquaponics solution, and the fourth system (D) is the aquaponics system (the sole source of nutrients is fish waste). By measuring pH (6.9) and EC (0.1 dSm^−1^), the level of nutrients was kept constant [1]. Systems A, B, and C were prepared using tub cultures, and ten seedlings were placed in each system.

### 2.2. Calibration of SPAD-502 Readings for Chlorophyll Assessment 

Estimating chlorophyll using direct laboratory methods (solvent extraction and then spectrophotometric estimation) is highly accurate. However, it has some limitations, such as precluding tracking dynamic changes in the chlorophyll content of plants as well as the lack of real-time detection. Therefore, the SPAD-502 m was used in this study to determine chlorophyll as it is a non-destructive, user friendly, inexpensive, early detection, and instant method in real time [40]. Several studies have revealed that SPAD-502 readings strongly indicate the chlorophyll content in plant leaves [8,9,10,11,12,13]. Hence, it has been used in many studies to estimate chlorophyll content in crops [14,15,16,17,18,19,20,21,22]. To evaluate the SPAD-502 device in estimating chlorophyll content and to confirm the reliability of these measurements in real time, a set of 85 plant leaves was first scanned using the SPAD-502 m. The leaves were then picked, labeled, and transferred to the laboratory for chemical assessment of chlorophyll content. Briefly, the sample (1 g) was ground, homogenized with 20 mL of concentrated acetone, filtered, and adjusted to 50 mL of 80% acetone. Absorbance (A) readings were recorded at 645 and 663 nm [9,10] using a DR 5000 UV spectrophotometer (HACH, London, ON, Canada). Chlorophyll content was then calculated using the following equation [9]: (1)$$Total\;chlorophyll=\left(20.2\times A_{645}\right)+\left(8.02\times A_{663}\right)$$
where $A_{645}$ is the absorbance at 645 nm and $A_{663}$ is the absorbance at 663 nm.

A simple correlation relationship between values resulting from SPAD-502 and those determined from chemical assessment was built to validate the reliability of using SPAD-502 for the subsequent experiments. A total of 3600 SPAD-502 values were collected through the growing season. All plants were measured six times during the growing season, and three leaves were randomly selected from each plant. Five readings per leaf between the midrib and the leaf margin and at the central point of a leaf were acquired. To reduce the potential impact of light intensity on chloroplast mobility, SPAD-502 measurements were conducted between 7:00 and 9:00 a.m.

### 2.3. Spectral Dataset Collection and Preprocessing 

Using a full-range hyperspectral ASD FieldSpec 4 Hi-Res (Analytical Spectral Devices Inc., Boulder, CO, USA) spectroradiometer, 3600 hyperspectral reflectance measurements were acquired for lettuce under different nutrient levels. A fiber optic probe was used. The range of the ASD device is between 350 and 2500 nm, with a resolution ranging between 3 nm below 1000 nm and 10 nm between 1000 and 2500 nm. A white barium sulfate plate (Labsphere, Inc., North Sutton, NH, USA) was used to calibrate the device to avoid the influence of any changes in atmospheric conditions prior to the measurement process. Simultaneously with spectral measurements, SPAD-502 Plus (Konica Minolta Sensing, Osaka, Japan) was used to measure the chlorophyll content of leaves. The spectral data obtained using ASD are in the form of an unprocessable file (.asd), and ViewSpecPro 6.02 software was used to convert the data into a Microsoft Excel values file (.csv) format. The average spectral data (X-variable), as well as the corresponding SPAD-502 readings (Y-variable), were stored in an Excel file to perform different preprocessing treatments, such as standard normal variate (SNV), to reduce the data scattering in the near-infrared (NIR) zone [41], and a Savitzky–Golay filter was used to smooth the spectral data [42]. Then, the data were divided into training (70%) and testing (30%) sets using a random subsampling technique to provide accurate analysis and prevent data bias.

### 2.4. Spectral Vegetation Indices (SVIs) 

Spectral vegetation indices are derived from the establishment of a mathematical correlation between two or more wavelengths. The spectral response to physical, chemical, or biological properties differs across different plants. Consequently, these properties exhibit distinct spectral indices compared to other plants due to their varying responsiveness to different wavelengths [43]. The wavelengths associated with the plant’s nitrogen and chlorophyll contents may overlap, indicating that multiple properties of the plant might share the same wavelengths [43]. In this study, SVIs that are most sensitive to chlorophyll content were derived, as shown in Table 1. The construction of vegetation indices combines several sensitive bands of chlorophyll content to minimize the influence of the plant’s environmental background (e.g., non-vegetated target soil, water body, etc.) as much as possible.

### 2.5. Color Vegetation Indices (CVIs)

A near portable digital camera (PowerShot SX720 HS, Canon Inc., Tokyo, Japan) was used to acquire images from the experiment. A grand total of 800 raw digital images were collected from all plants in conjunction with SPAD-502 readings. These images contain a large amount of noise and non-vegetal objects; hence, they have been enhanced and segmented to separate them from the background [49]. In this study, a robust deep convolutional neural network architecture, SegNet, was used for the segmentation task (Figure 3) [50]. Red, green, and blue features were extracted from the segmented image dataset, and then the most sensitive vegetation indices for chlorophyll were derived [30]. The feasibility of these indices was evaluated for modeling plant chlorophyll content and compared with SVIs to adopt the best indices as reliable estimators of chlorophyll (Table 2).

### 2.6. Design of AutoML Models

The concept of ‘AutoML’ pertains to automating machine learning activities, thereby minimizing or eliminating the need for manual intervention [51]. AutoML has provided individuals without technical or subject experience with the ability to utilize machine learning techniques in addressing specific problems [52]. The primary objective of most AutoML is to achieve complete automation of model selection, hyperparameter optimization, and feature selection procedures [53]. Previously, several approaches and strategies focused on specific aspects of the AutoML process. However, a range of fully automated approaches have been developed in recent years [54,55,56,57]. The AutoML automated approach encompasses sequential procedures to prepare the selected model for prediction:

Model Selection: The primary aim of model selection is to determine the ML models that exhibit the highest level of accuracy when trained on a particular dataset [54]. AutoML aims to identify the best-fitted model for a given dataset without any human intervention. This is achieved by iteratively training multiple models on the same input data and picking the model with the highest performance [55]. 

Hyperparameter optimization (HPO): HPO is a crucial process in machine learning where the appropriate setting and adjustment of hyperparameters can significantly improve the performance of a model. Furthermore, previous studies have demonstrated that careful selection of hyperparameters significantly enhances models’ efficacy compared to default model configurations [56].

Feature engineering: This is a crucial phase in the machine learning process that can be effectively accomplished with AutoML. When conducted manually, this task can be laborious and monotonous [55].

Recently, there has been a proliferation of frameworks that aim to integrate the three preceding steps of AutoML. Some examples of automated machine learning frameworks are EvalML AutoKeras, AutoGluon, Auto-Weka, and Auto-PyTorch [57], among others. EvalML is an open-source AutoML framework that facilitates the automated execution of various tasks, such as feature selection, model selection, and hyperparameter optimization. The EvalML routine employs a random forest classifier/regression for feature selection and Bayesian optimization to optimize the hyperparameters of the pipeline. EvalML constructs and optimizes machine learning pipelines based on a specified objective function parameter, such as mean squared error (MSE), for time series prediction. It supports a range of supervised machine learning problems, encompassing regression, classification, time series regression, and time series classification. The problem presented in this work is to predict the chlorophyll content of lettuce leaves grown in aquaponics systems. Therefore, EvalML is employed to search for optimal models and optimize the acquired spectral data. It has been proven that random searching for hyperparameters is more effective and accurate than grid search for developing AutoML [58]. Therefore, this study used random search to determine the model architecture, optimizer, and learning rate to obtain the highest prediction accuracy. If the model’s accuracy is unsatisfactory, another architecture with new hyperparameters is employed for retraining. A summary of the major components of the AutoML pipeline is illustrated in Figure 4 [59].

To develop a predictive model for the chlorophyll content of aquaponically grown lettuce using AutoML, SPAD-502 readings were collected in conjunction with the acquisition of the spectral and image datasets. Then, the SVIs and CVIs most closely related to plant chlorophyll content were calculated. These indices were used as inputs to the open-source AutoML framework (EvalML). The AutoML procedures were implemented using Python on Google Collaboratory through its website. The datasets for SVIs and CVIs were then loaded separately and combined. The data were cleaned using EvalML’s DefaultDataChecks for validation, as EvalML accepts a Pandas data frame as input. The data-checking process also includes a built-in function to validate the data by checking for errors and recommending preprocessing. This study suggested automated data preprocessing using the lognormal transformation to apply the lognormal transformation to the data as a normalization and preprocessing procedure. To divide the data into training and testing sets, the AutoSplit function of EvalML was used, where the dataset was automatically divided into a 70% training dataset and a 30% testing dataset. The AutoML search function of the EvalML framework was applied to allow the machine to identify the best prediction models by passing the training data, the type of problem, and the number of batches, which returned the top predictive models to model the data. Four predictive models that were best for modeling all datasets for this study were given: extra trees (ETs), LightGBM (LGBM), XGBoost (XGB), and random forest (RF). Table 3 shows the EvalML framework’s automatically best selected hyperparameters. All the models contained an imputer for replacing missing data.

### 2.7. Performance Evaluation of the Regression Models

Evaluating the precision of predictive algorithms is the final stage, emphasizing the prediction capabilities of the proposed models. Performance evaluation of the models involves establishing a correlation between the measured and expected values and subsequently calculating performance metrics. Two core metrics that highlight the model’s predictive power are the coefficient of determination ($R^{2}$) (Equation (2)) and root mean square errors (RMSEs) (Equation (3)) for the calibration ($R_{c}^{2},\;RMSE_{c}$) and prediction ($R_{p}^{2},\;RMSE_{p}$) datasets. The maximum $R^{2}$ and minimum RMSE indicate the best model. The best four machine learning algorithms—back-propagation neural network (BPNN), partial least squares regression (PLSR), random forest (RF), and support vector machine (SVM)—were used as a comparison benchmark and to evaluate the performance of the AutoML system. The data split was 70% (2520 samples) for training and 30% (1080 samples) for testing. The open-source program HSI-PP V1.2 was used to perform preprocessing, waveband selection, and statistical analyses for BPNN, PLSR, RF, and SVM algorithms [60]. To further investigate the performance of AutoML, the Regression Learner (R_leraner_) application built into MATLAB R2022b (The MathWorks, Inc., Natick, MA, USA) was used to train all the traditional machine learning algorithms (29 algorithms) as a benchmark, as well.
(2)$$RMSE=\sqrt{\frac{1}{n}\overset{n}{\underset{j=1}{\sum }}{\left(y_{j}-y_{p}\right)}^{2}}$$
(3)$$R^{2}=1-\frac{\sum_{j=1}^{n}{\left(y_{j}-y_{p}\right)}^{2}}{\sum_{j=1}^{n}{\left(y_{j}-y_{m}\right)}^{2}}$$
where, $y_{j}$, $y_{p}$, $y_{m}$ are measured, predicted, and mean of measured chlorophyll content for sample *j*, and *n* is the number of samples in the dataset.

In this study, the computed SVIs and CVIs values were taken to be the predictive variable (X) and the SPAD-502 readings (chlorophyll content) to be the response variable (Y). The predicted values of chlorophyll were computed with Equation (4) [61]: (4)$$Chlorophyll=\overset{n}{\underset{i=1}{\sum }}\beta_{i}H_{i}+C$$
where $\beta_{i}$ is the fit model coefficient of the models, $H_{i}$ is the spectrum of each pixel in the spectral data, and *C* is constant. 

## 3. Results and Discussion

### 3.1. Efficacy of SPAD-502 Values for Chlorophyl Content Estimation

The relationship between chlorophyll contents in plant leaves estimated by the laboratory wet chemistry assessment method and those resulting from SPAD-502 values is shown in Figure 5. High linear correlations were obtained between SPAD-502 readings and chlorophyll content, with $R^{2}$ = 0.95 and the correlation coefficient (r) = 0.975. Many scientific contributions support this result. For instance, Mendoza-Tafolla et al. studied the relationship between chlorophyll content and SPAD-502 values of romaine lettuce leaves, indicating a strong relationship with $R^{2}$ = 0.97 and r = 0.99 [9]. In addition, Jiang et al. found a significant correlation between chlorophyll content readings and SPAD-502 values for tomato leaves [10]. Uddling et al. evaluated the relationship between leaf chlorophyll concentration and SPAD-502 m readings for birch and wheat plants and found a significant correlation between them with an $R^{2}$ of more than 0.90 [11]. Moreover, Wakiyama assessed the relationship between SPAD-502 values and rice’s chlorophyll content, indicating a strong correlation with $R^{2}$ = 0.94 [12]. Xiong et al. also studied the relationship between chlorophyll content and SPAD-502 readings for tomatoes and zizania, noting a strong relationship with r of 0.90 and 0.97, respectively [13]. A strong positive relationship between SPAD-502 readings and chlorophyll content in wheat with $R^{2}$ = 0.93 was reported by Shah et al. [8]. SPAD-502 values have therefore been widely used for estimation of crop chlorophyll content and for guidance of plant health status and topdressing [14,15,16,17,18,19,20,21,22]. Consequently, this motivated its use in this study to estimate total chlorophyll in aquaponically grown lettuce.

### 3.2. Spectral Vegetation Indices

The evaluation results of the AutoML and the comparative machine learning models (BPNN, RF, PLSR, SVM, and R_learner_ algorithm) are tabulated in Table 4. The most fitted models have been shown in Figure 6, Figure 7 and Figure 8, including the best-selected SVIs, coefficients of determination ($R^{2}$), and root mean square errors (RMSEs) for the calibration ($R_{c}^{2},\;RMSE_{c}$) and prediction ($R_{p}^{2},\;RMSE_{p}$) datasets. A total of nineteen vegetation indices, which are highly responsive to the chlorophyll content of plants, were chosen to create a reliable model for predicting chlorophyll content. In general, the results showed that the AutoML system performed better than traditional models for all vegetative indices. Of all vegetation indices, GRVI achieved the best prediction results and is, therefore, the best index for modeling chlorophyll content. For GRVI, the AutoML achieved the highest predictive accuracy with $R_{p}^{2}=0.91$, while RF, PLSR, BPNN, SVM, and Gaussian process regression (R_Learner_) obtained $R_{p}^{2}$ of 0.89, 0.82, 0.82, 0.81, and 0.81, respectively, as shown in Table 4 and Figure 6a–c. These findings led us to conclude that the GRVI can be safely used as an accurate estimator of the chlorophyll content of plant leaves. The superiority of the GRVI can be attributed to its strong correlation with the nitrogen content of plants, a crucial structural element of chlorophyll [62]. Additionally, compared to many other vegetation indices, the GRVI has reportedly been more sensitive (red-based) to chlorophyll content [63]. These findings align with those of Maresma et al. who utilized the GRVI index to estimate maize’s nitrogen content (as it is the main structural component of chlorophyll) which outperformed all other vegetation indices [64].

Regarding the CIgreen, it is also characterized by its close relationship with plant chlorophyll [48]. It has been reported to be a very good estimator of the chlorophyll content of leaves [48]. AutoML achieved a high level of predictive accuracy, with an $R_{p}^{2}$ of 0.90, outperforming RF, PLSR, BPNN, SVM, and Gaussian process regression (RLearner), which achieved $R_{p}^{2}$ values of 0.85, 0.82, 0.87, 0.81, and 0.83, respectively, as shown in Table 4 and Figure 6d–f. These results are consistent with Clevers et al. (2017), who used the CIgreen index to estimate leaf chlorophyll and achieved high predictive accuracy of $R_{p}^{2}$ ranging from 0.81 to 0.90 [48]. The advantage of GRVI and CIgreen over their counterparts as accurate estimators of plant chlorophyll may be because the wavelengths (559–872 nm) used to calculate these indices fall within the range of influence of nitrogen and chlorophyll [48].

NDVIs are also classified as reliable indices associated with chlorophyll content [21]. Moreover, NDVI measures an area’s greenness, indicating the leaves’ chlorophyll content. However, it was outperformed by GRVI and CIgreen. The NDVI achieved good predictive performance with an $R_{p}^{2}$ of 0.89, 0.84, 0.79, 0.78, 0.77, and 0.83 for AutoML, BPNN, PLSR, RF, SVM, and Ensemble, respectively, as illustrated in Table 4 and shown in Figure 6g–i. These results are consistent with Narmilan et al., who used NDVI and GNDVI as estimators for chlorophyll content in Sugarcane, achieving good prediction accuracy with $R_{p}^{2}$ of 0.82 and 0.86 [21]. Hence, our study achieved higher prediction accuracy using AutoML, which proves its superiority over traditional machine learning algorithms.

The MCARI index quantifies the extent of chlorophyll absorption and is responsive to alterations in the structural composition of chlorophyll and fluctuations in the leaf area index (LAI) [3]. Notably, MCARI values are not affected by the environmental lighting conditions surrounding the plant [3]. MCARI index did well in estimating chlorophyll, with an $R_{p}^{2}$ of 0.88, 0.85, 0.82, 0.78, and 0.84 for AutoML, BPNN, RF, PLSR, and Gaussian process regression, respectively, as shown in Figure 7a–c. The outstanding predictive ability of MCARI with AutoML in chlorophyll estimation is due to its sensitivity to chlorophyll content and its ability to predict plant nitrogen content [65]. This result agrees with Wu et al., who used the MCARI index to estimate the chlorophyll content in maize leaves, achieving good regression accuracy with correlations $R_{p}^{2}$ of 0.88. They stated that the MCARI index is more efficient than SR index because the latter considers the effect of the leaf area index (LAI) [3]. 

The VREI results were also interesting, especially with AutoML achieving very good predictive accuracy with an $R_{p}^{2}$ of 0.87. Moreover, it performed well with traditional algorithms, obtaining $R_{p}^{2}$ of 0.83, 0.78, 0.77, 0.77, and 0.78 for BPNN, PLSR, RF, SVM, and Ensemble, respectively, as illustrated in Table 4 and shown in Figure 7d–f. Our findings are consistent with those of Velichkova and Krezhova, who estimated chlorophyll in pepper plants using VREI, achieving very good predictive accuracy [45]. The reliability of the VREI as an estimator of chlorophyll is due to its sensitivity to the combined effects of foliage chlorophyll concentration, canopy leaf area, and water content [45].

MSR1 performed well in estimating and predicting the chlorophyll content of plant leaves, also achieving outstanding prediction accuracy using AutoML with an $R_{p}^{2}$ of 0.86, superior to traditional methods, which obtained $R_{p}^{2}$ of 0.78, 0.73, 0.72, 0.68, and 0.73 for BPNN, PLSR, SVM, RF, and Gaussian, respectively, as illustrated in Table 4 and shown in Figure 7g–i. These results are in line with the study conducted by Haboudane et al., who used MSR to estimate chlorophyll, achieving good predictive quality with an $R_{p}^{2}$ of 0.80 [66]. Our study is also consistent with El-Hendawy et al. [46], who used MSR to estimate chlorophyll in wheat, obtaining $R_{p}^{2}$ of 0.65 and 0.73. MSR is a good estimator of chlorophyll because it is considered more linearly related to vegetation parameters. In addition, MSR and SR are the most affected by chlorophyll variability, showing high sensitivity even at high chlorophyll levels (up to 60 µg/cm^2^) [66]. 

All vegetation indices were combined into a single input dataset to further investigate the performance of AutoML for predicting chlorophyll content using vegetation indices. Hence, all calculated SVIs were transferred to one file to represent the predictive variables (X-data), while the chlorophyll content estimated from the SPAD-502 device was used as the Y variable. The combination of such indices had a stimulating effect on the prediction efficiency of the AutoML system and the other machine learning models. In general, the AutoML model (XGBoost Regressor) significantly outperformed the manual machine learning models by achieving very high prediction accuracy with a coefficient of determination ($R_{p}^{2}$) of 0.93. Manual machine learning models also achieved good prediction accuracy with $R_{p}^{2}$ of 0.91, 0.89, 0.87, 0.85, and 0.83 for BPNN, RF, PLSR, SVM, and Ensemble, respectively, as shown in Figure 8. These results are consistent with the study of Haboudane et al. (2004), who combined a series of vegetation indices divided into three categories (NDVI, MSR, and MCARI, TCARI) and the integrated forms (MCARI/OSAVI and TCARI/OSAVI)) to estimate the chlorophyll content of winter wheat. Their study achieved high estimation accuracy with an $R_{p}^{2}$ of 0.94 [67].

### 3.3. Color Vegetation Indices

The outcomes of assessing CVIs as estimators of chlorophyll are displayed in Table 5, while the best-performed models are showcased in Figure 9. IKAW demonstrated superior prediction performance using AutoML, confirming it as the most dependable index when compared to other CVIs [28,30]. The IKAW index achieved good predictive accuracy with an $R_{p}^{2}$ of 0.85, as shown in Table 5 and Figure 9a. The performance of the other models, including RF, PLSR, BPNN, SVM, and RLearner, was comparatively inadequate when compared to AutoML. These results agree with those reported by many scientific contributions that have proven that the IKAW ((R − B)/(R + B)) index is the most fitted function of RGB space to estimate the chlorophyll content of leaves [28,30,68,69]. Mahmoodi et al. used the IKAW index as an estimator of chlorophyll and demonstrated a strong correlation between the index and leaf chlorophyll content, achieving good estimation accuracy with a correlation coefficient of 0.87 [30]. Also, reasonable prediction accuracy ($R_{p}^{2}$ = 0.82) was obtained for NDI under AutoML, as shown in Figure 9b. In this context, our results agree with Manuel and Blanco, who used the NDI as an estimator for chlorophyll, achieving good predictive quality with an $R_{p}^{2}$ of 0.9 [70]. GLI index came in third place in terms of the quality of color vegetation indicators as estimators of chlorophyll, obtaining $R_{p}^{2}$ of 0.81 developed under AutoML modeling, as shown in Figure 9c and Table 5. This result aligns with the findings of Saberioon et al., who employed the GLI index to assess the chlorophyll levels in rice plants. Their study demonstrated a reasonable correlation between GLI and chlorophyll content with an $R_{p}^{2}$ of 0.64 [71]. The GLI’s reasonableness in estimating chlorophyll content may be due to its association with the visible green band [71].

Figure 9d–i displays the rn, gn, and bn indices. The findings demonstrate a strong association between the gn index and the chlorophyll concentration. Conversely, the rn and bn variables exhibited weak correlations with the chlorophyll concentration, suggesting that the green band is more influential than the red and blue bands in influencing the chlorophyll content in crops [28]. The AutoML and BPNN models achieved $R_{p}^{2}$ values of 0.79 and 0.63, respectively, for the index ‘gn,’ as shown in Figure 9e,h. These results agree with those outlined by Zhang et al., who used the rn, gn, and bn indices to estimate chlorophyll in sorghum, achieving reasonable predictive quality with $R_{p}^{2}$ of 0.56, 0.64, and 0.48 for rn, gn, and bn, respectively [28]. The results revealed that for GRRI, RBRI, GBRI, and WI, their performance is insufficient, with $R_{p}^{2}$ ranging from 0.45 to 0.50 under AutoML modeling, as shown in Table 5.

To further investigate the reliability of CVIs as estimators of plant chlorophyll content, all CVIs were combined as one dataset, representing the independent variable, while chlorophyll content (SPAD-502 reading) defined the predictor variable. In general, combining indices gave more promising results than using each index separately. The results displayed in Figure 10 indicated that the developed AutoML (XGBoost Regressor) obtained the best predictive results, with the highest values for $R_{p}^{2}$ (0.85), outperforming all other traditional models. The BPNN model outperformed its counterparts by obtaining an $R_{p}^{2}$ of 0.78, and the RF, PLSR, SVM, and R_Learner_ model gave $R_{p}^{2}$ of 0.74, 0.73, 0.70, and 0.67, respectively. It is noted from these results that combining color indices has enhanced predictive accuracy compared to using indices individually. This is because most indices are linked to the green band and are mutually reinforcing when combined as a single input. Furthermore, studies have shown that CVIs associated with the green band significantly correlate with nitrogen and chlorophyll content [72,73]. Additionally, this can be attributed to the fact that the combination of different features and different bands provides an expanded and complete description of the target process [74]. Moreover, it indicates that JPEG color images captured by digital cameras with built-in enhancement functions are sensitive to the green band [72].

### 3.4. Fusion of SVIs and CVIs

This section evaluated the combination of SVIs and CVIs to predict chlorophyll content in aquaponically grown lettuce. The results indicated that the linear combination of SVIs and CVIs strongly correlated with chlorophyll content. Moreover, all regression models gave the best predictive accuracy with highest $R_{p}^{2}$ values. The combination of the two types of vegetation indices achieved very high prediction accuracy with an $R_{p}^{2}$ of 0.98 using the developed AutoML (XGBoost Regressor). In addition, all traditional models obtained high $R_{p}^{2}$ values, estimated at 0.96, 0.94, 0.93, 0.90, and 0.89 for BPNN, RF, PLSR, SVM, and R_Learner_ model, respectively, as shown in Figure 11. These results are consistent with the study by Zhang et al., who demonstrated that the linear combination of CVIs (red, green, and blue) and SVIs indicates the best predictive model for chlorophyll in sorghum crops [28]. As a result of incorporating types of vegetation indices, they achieved high prediction accuracy with $R_{p}^{2}$ = 0.90. The promised results of the combination may be because it has created a bridge between the genotype and phenotype of plants [28]. The promising results of the fusion of the SVIs and CVIs can be attributed to the fact that the effective combination of the sensors (a digital camera and a spectroradiometer) and the various features may lead to the expansion of individual sensor capabilities and the provision of a robust and complete description of an environment or targeted process rather than using an individual source alone [74]. Color images focus more on phenotypic traits, while spectral data look at plant physiology. The advantage of spectroscopic measurements over colored images lies in their ability to detect disturbances such as nutrient stress, water stress, or disease before the plant becomes symptomatic. In contrast, colored images cannot detect these disturbances until the plant shows visible symptoms [75]. Hence, the combination gave a comprehensive and reliable idea about the condition of the plant in general and chlorophyll in particular due to the derivation of the associated indices. 

Upon careful evaluation of prior research that employed conventional machine learning techniques to model chlorophyll content, we deduce that AutoML systems exhibit superiority. In the study of Shah et al. (2019), the authors used a random forest regression algorithm with 45 spectral VIs to predict the wheat leaves’ chlorophyll content, obtaining a high prediction accuracy with an $R_{p}^{2}$ of 0.89 [2]. Using fifteen vegetation indices, Narmilan et al. used machine learning regression models along with SPAD-502 measurements to find out how much chlorophyll was in sugarcane crops. It was discovered that RF performed better than other models, with an $R_{p}^{2}$ = 0.92 [21]. They used the algorithms before and after selecting the optimal wavebands, with the former training them on all 24 vegetation indices across five spectral bands and the latter training them on 15 indices. In the same context, Ta et al. estimated the content of the apple leaves from chlorophyll using several learning algorithms: (1) univariate linear regression (ULR); (2) multivariate linear regression (MLR); (3) SVMR; and (4) RF using the SVIs and the readings of SPAD-502 as the response variable. Their results demonstrated that the random forest outperformed all proposed algorithms, achieving the highest predictive accuracy with $R_{p}^{2}$ = 0.93 and RMSE = 0.95 [20]. An et al. proposed a predictive framework to estimate chlorophyll content in rice using regression models (GPR, RF, SVM, and GBR). All the algorithms performed well and the random forest surpassed the best prediction performance with RMSE = 1.54 and $R_{p}^{2}$ = 0.92 [22]. Our study is consistent with that conducted by Yi-Cheng Huang et al. compared to AutoML, VGG-16, ResNet-50, and MobileNet v1 for defect detection on cylindrical metal surfaces, where it was proven that AutoML outperformed other models by obtaining an $R_{p}^{2}$ of 0.9983 [67]. Furthermore, our results are consistent with those achieved by Borja Espejo-Garcia et al., where AutoML was used to distinguish between plants and weeds, achieving an identification quality of 93.8% [33]. Zhang et al. used vegetation indices derived from RGB, hyperspectral, and fluorescence images using the PLSR model, achieving good predictive accuracy with an $R_{p}^{2}$ that ranged from 0.67 to 0.88 when using each type independently, while the $R_{p}^{2}$ value increased to 0.90 when combined [28]. Since chlorophyll is essential for precision agriculture as it is the vital pigment in photosynthesis. Furthermore, it is a good indicator of mutations, salinity, drought stress, and nutritional status [3,76]. Understanding the chlorophyll content of plants is crucial for guiding plant cultivation management. In this context, therefore, this study is an important scientific contribution. The superiority of AutoML systems over manual machine learning approaches depends on the automatic ability of these systems to select the best models and their related hyperparameters (such as the learning rate and optimizer) and features, as well as the parameters related to the model architecture, such as the number of layers and operation of each layer [53]. This process is repeated automatically using many algorithms to reach the best scenarios to model the target task with the best experimental results without or with minimal programming knowledge of the user [67]. In contrast, traditional machine learning models, which rely on manually selecting models and hyperparameters based on the user’s point of view and software experience, may often be flawed, leading to low model accuracy.

## 4. Conclusions

This study investigated the feasibility of developing an automated machine learning (AutoML) pipeline specifically to develop chlorophyll estimation models in aquaponically grown lettuce using SVIs and CVIs. The AutoML and regression models (BPNN, PLSR, RF, and SVM) were validated using 3600 hyperspectral measurements and 800 RGB images. SVIs and CVIs were derived from spectral image datasets. The chlorophyll content was measured using an SPAD-502 m calibrated by laboratory analysis. The results revealed a strong positive relationship between chlorophyll content and SPAD-502 readings, with an $R^{2}$ of 0.95. Furthermore, the AutoML system generated four optimal models (extra trees, LightGBM, XGBoost, and random forest) selected for chlorophyll modeling. The results demonstrated that the developed AutoML models outperform all traditional models by obtaining the highest values of $R_{p}^{2}$ for all vegetation indices. GRVI achieved the highest predictive accuracy ($R_{p}^{2}$ of 0.91), outperforming other spectral indices. Similarly, IKAW outperformed other color indices by obtaining an $R_{p}^{2}$ of 0.85. The combination of spectral and color vegetation indices achieved the best predictive accuracy and the highest $R_{p}^{2}$ values, which reached 0.98 with the developed AutoML model (XGBoost), outperforming all traditional models. Due to the flexibility of the proposed methodology, it can also be used with minor modifications to estimate chlorophyll content in not only greenhouse crops but also field crops such as wheat, corn, and barley. Furthermore, our future endeavors will focus on designing fully automated aquaponics systems, where the developed model will be integrated and evaluated for its practical viability, thus bridging the gap between theoretical research and real-world application.

## Figures and Tables

**Figure 1 plants-13-00392-f001:**
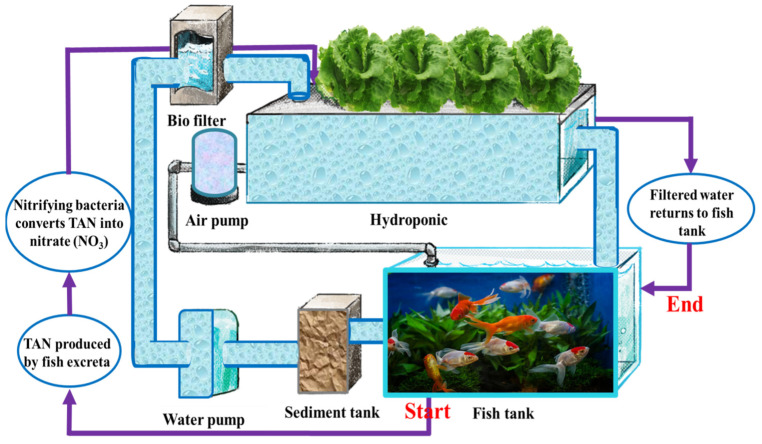
Schematic diagram of aquaponics system.

**Figure 2 plants-13-00392-f002:**
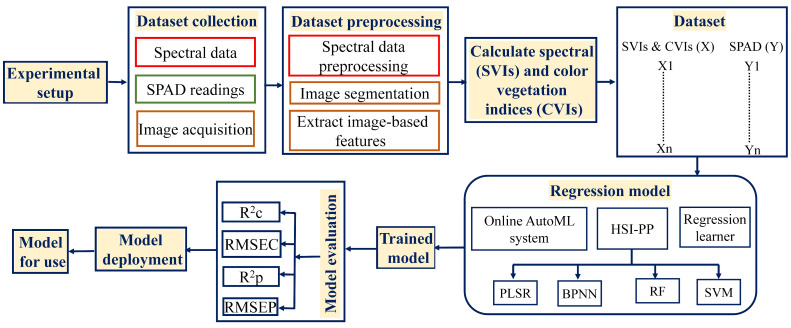
The proposed method for chlorophyll estimation in aquaponics plants.

**Figure 3 plants-13-00392-f003:**
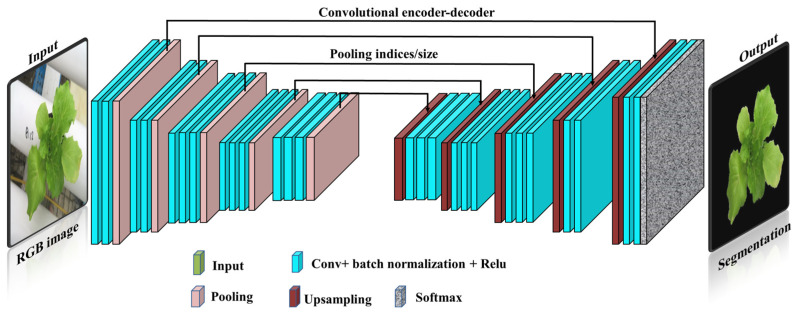
An illustration of the deep convolutional neural network SegNet architecture for image segmentation used to isolate plants from the background.

**Figure 4 plants-13-00392-f004:**
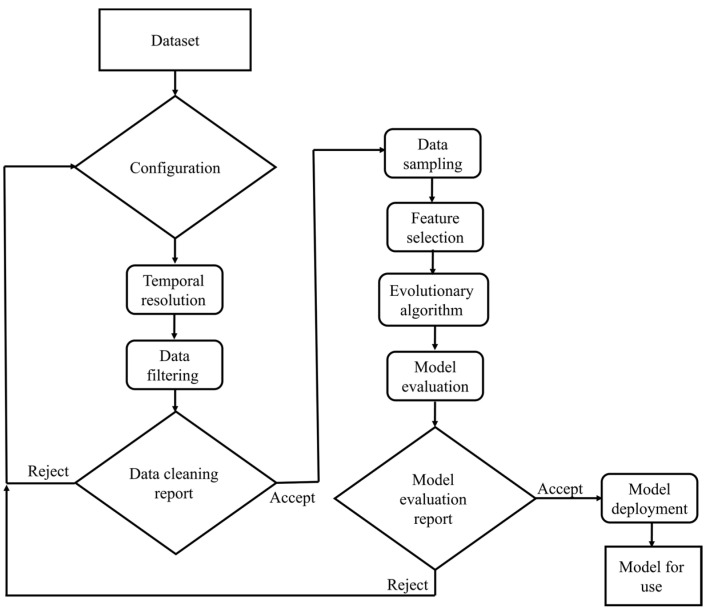
Summary of the major components of the AutoML pipeline.

**Figure 5 plants-13-00392-f005:**
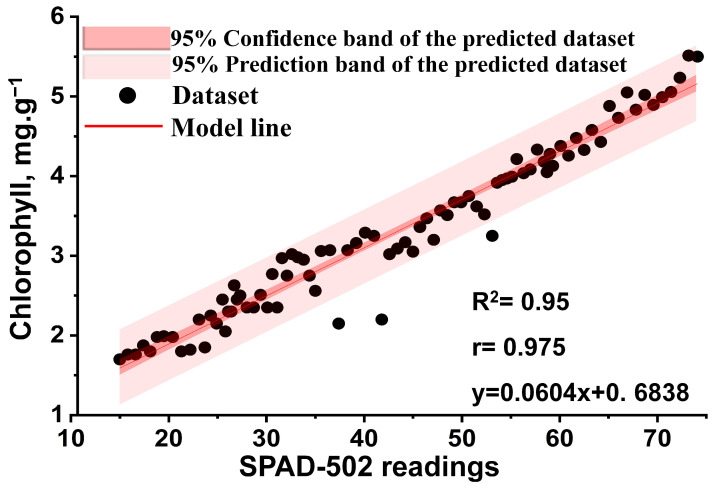
Relationship between chlorophyll content and SPAD-502 readings at confidence interval (α) of 0.05.

**Figure 6 plants-13-00392-f006:**
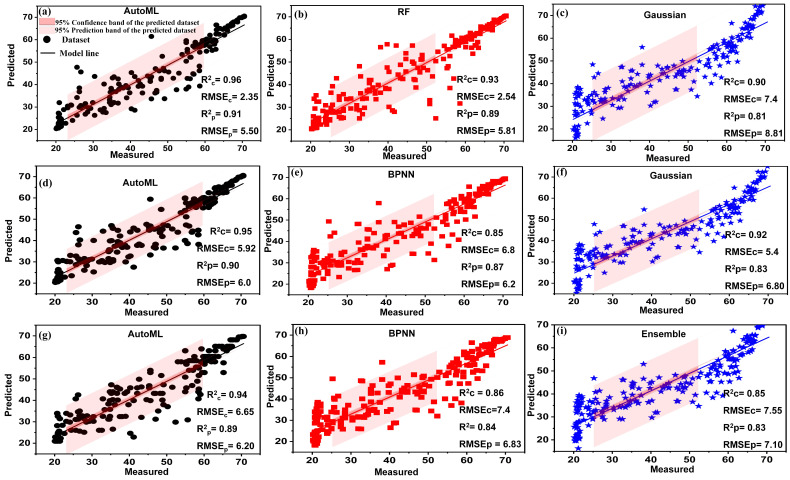
Relationship between measured and predicted values of chlorophyll content estimation using GRVI (**a**–**c**), CIgreen (**d**–**f**), and NDVI (**g**–**i**) for AutoML and comparative models at α = 0.05.

**Figure 7 plants-13-00392-f007:**
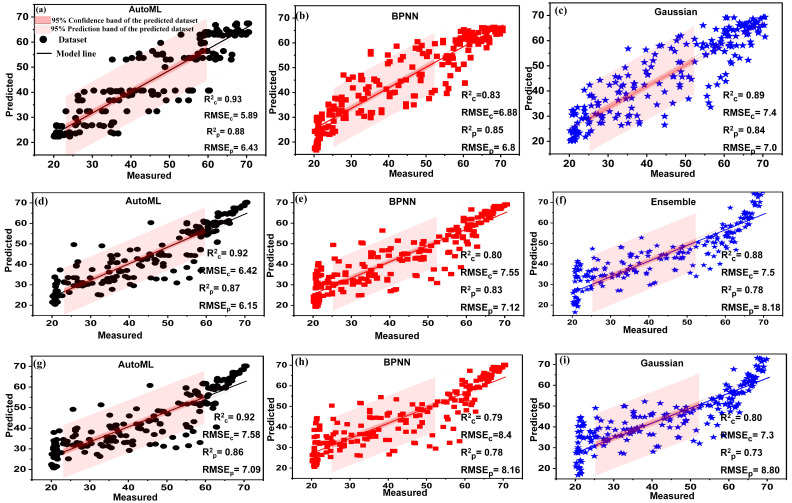
Relationship between measured and predicted values of chlorophyll content estimation using MCARI (**a**–**c**), VREI (**d**–**f**), and MSR1 (**g**–**i**) for AutoML and comparative models at α = 0.05.

**Figure 8 plants-13-00392-f008:**
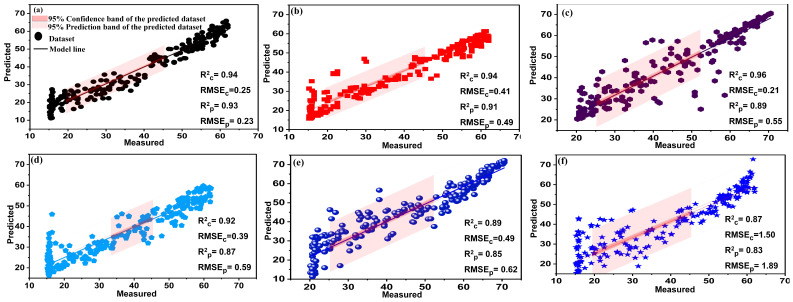
The relationship between measured and predicted values of chlorophyll for all trained models. AutoML (**a**), BPNN (**b**), RF (**c**), PLSR (**d**), SVM (**e**), Ensemble (**f**) using all SVIs at α = 0.05.

**Figure 9 plants-13-00392-f009:**
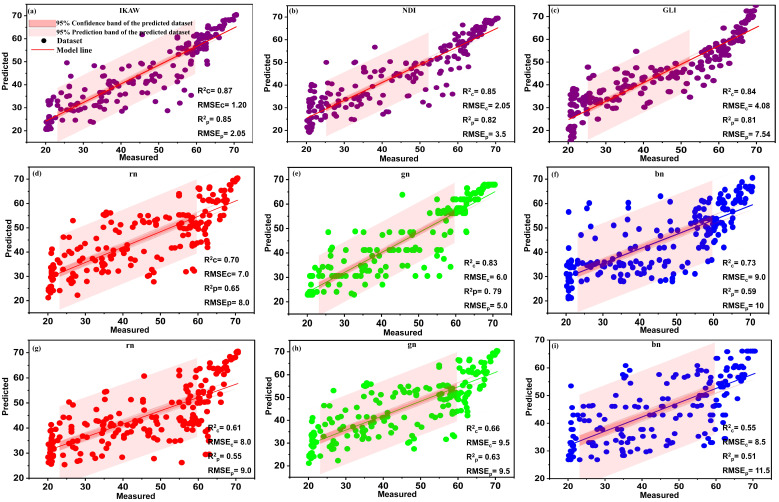
The relationship between measured and predicted values of different CVIs as predictors of chlorophyll content using AutoML models (**a**–**f**) and BPNN models (**g**–**i**) at α = 0.05.

**Figure 10 plants-13-00392-f010:**
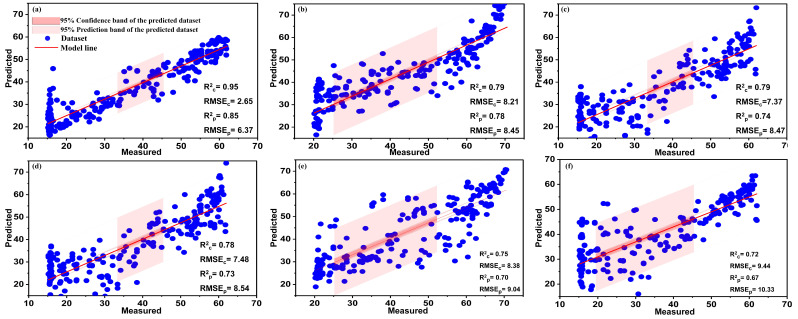
The relationship between measured and predicted values of chlorophyll content for AutoML (**a**), BPNN (**b**), RF (**c**), PLSR, (**d**), SVM (**e**), and R_learner_ model (**f**) using the combination of CVIs at α = 0.05.

**Figure 11 plants-13-00392-f011:**
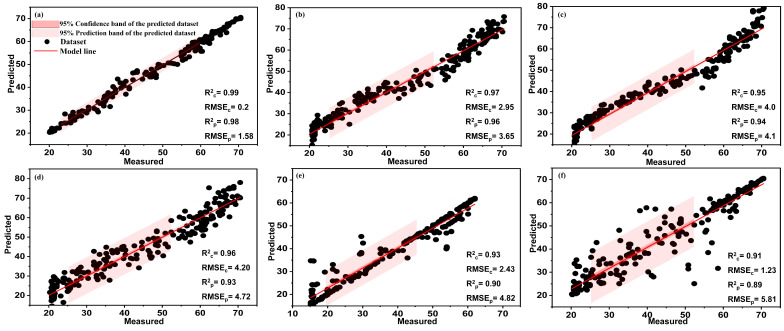
The relationship between measured and predicted values of chlorophyll content for AutoML (**a**), BPNN (**b**), RF (**c**), PLSR, (**d**), SVM (**e**), and R_learner_ model (**f**) using the combination of SVIs and CVIs at α = 0.05.

**Table 1 plants-13-00392-t001:** SVIs for estimating chlorophyll content in aquaponically grown lettuce.

No.	VIs	Formula	Ref.
1	Normalized Difference Vegetation Index (NDVI)	(R750 − R705)/(R750 + R705)	[6]
2	Normalized Difference Vegetation Index-1 (NDVI1)	(R750 − R680)/(R750 + R680)	[6]
3	Normalized Difference Vegetation Index-3D (NDVI3D)	(R780 − R715)/(R780 + R715)	[6]
4	Vogelmann Red Edge Index2 (VREI 2)	(R740/R720)	[44]
5	Modified simple ratio of reflectance-1 (MSR1)	(R750 − R445)/(R705 − R445)	[6]
6	Structure insensitive pigment index (SIPI)	(R800 − R445)/(R800 − R680)	[2]
7	Modified Datt index (MDATT1)	(R703 − R732)/(R703 − R722)	[3]
8	Modified Chlorophyll Absorption Ratio Index (MCARI)	[(R702 − R671) − 0.2 × (R702 − 549)] × (R702/R671)]	[6]
9	Green Ratio Vegetation Index (GRVI)	R872/R559	[21]
10	Photochemical Reflectance Index (PRI)	(R531 − R570)/(R531 + R570)	[6]
11	Visible Atmospherically Resistant Index (VARI)	(R559 − R661/R559 + R661 − R488)	[2]
12	Vogelmann Red Edge Index (VREI)	(R740/R720)	[45]
13	Simple Ratio Index (SR)	(R810/R550)	[6]
14	Red Edge Vegetation Stress Index (RVSI)	(0.5(R722 + R763) − R733)	[46]
15	Blue/Green pigment Index-1 (BGI1)	(R450/R550)	[6]
16	Lichtenthaler index 2 (Lic2)	(R790 − R680)/(R790 + R680)	[47]
17	Plant Senescence Reflectance Index (PSRI)	((R680 − R500)/R750)	[6]
18	Normalized Pigment Chlorophyll Index (NPCI)	(R642-R432)/(R642 + R432)	[20]
19	Green chlorophyll index (CIgreen)	(R780/R550) − 1	[48]

**Table 2 plants-13-00392-t002:** The RGB vegetation indices are used to estimate the chlorophyll content.

No.	RGB Index	Formula	Ref.
1	Normalized red index (rn)	R/(R + G + B)	[29]
2	Normalized green index (gn)	G/(R + G + B)	[29]
3	Normalized blue index (bn)	B/(R + G + B)	[29]
4	Green, red ratio index (GRRI)	G/R	[18]
5	Red, blue ratio index (RBRI)	R/B	[29]
6	Green, blue ratio index (GBRI)	G/B	[18]
7	Kawashima index (IKAW)	(R − B)/(R + B)	[30]
8	Normalized difference index (NDI)	(rn − gn)/(rn + gn + 0.01)	[29]
9	Woebbecke index (WI)	(G − B)/(R − G)	[18]
10	Green leaf index (GLI)	(2G − R − B)/(2G + R + B)	[18]

**Table 3 plants-13-00392-t003:** The EvalML framework’s automatically best selected hyperparameters.

Index	Pipeline Name	Hyperparameters
0	ET	{’categorical_impute_strategy’: most_frequent, ’numeric_ impute_strategy’: median, ’booleane_impute_strategy’: most_frequent, ’categorical_fill_value’: None, ’numeric_ fill_value’: None, ’booleane_ fill_value’: None}, ‘Extra Trees Regressor’: {’n_estimators’: 100, ‘max_features’: ‘auto’, ‘max_depth’: 6, ‘min_samples_split’: 2, ‘min_weight_fraction_leaf’: 0.0, ‘n_jobs’: −1}}
1	XGB	{’categorical_impute_strategy’: most_frequent, ’numeric_ impute_strategy’: median, ’booleane_impute_strategy’: most_frequent, ’categorical_fill_value’: None, ’numeric_ fill_value’: None, ’booleane_ fill_value’: None}, ‘XGBoost Regressor’: {’eta’: 0.1, ‘max_depth’: 6, ‘min_child_weight’: 1, ‘n_estimators’: 100, ‘n_jobs’: −1}}
2	LGBM	{’categorical_impute_strategy’: most_frequent, ’numeric_ impute_strategy’: median, ’booleane_impute_strategy’: most_frequent, ’categorical_fill_value’: None, ’numeric_ fill_value’: None, ’booleane_ fill_value’: None}, ‘LightGBM Regressor’: {boosting_type: gbdt, learning_rate: 0.1, n_estimators: 20, max_depth’: 0, ‘num_leaves’: 31, Win_child_samples’: 20, ‘n-jobs’: −1, ‘bagging_freq’: 0, ‘bagging_fraction’: 0.9}}
3	RF	{’categorical_impute_strategy’: most_frequent, ’numeric_ impute_strategy’: median, ’booleane_impute_strategy’: most_frequent, ’categorical_fill_value’: None, ’numeric_ fill_value’: None, ’booleane_ fill_value’: None}, ‘Random Forest Regressor’: {’n_estimators’: 482, ‘max_depth’: 25, ‘n_jobs’: −1}}

**Table 4 plants-13-00392-t004:** The SVIs are used to estimate the chlorophyll content in aquaponically grown lettuce.

VIs	AutoML			RF		PLSR		BPNN		SVM		R_Learner_	
	Model	$R_{p}^{2}$	$RMSE_{p}$	$R_{p}^{2}$	$RMSE_{p}$	$R_{p}^{2}$	$RMSE_{p}$	$R_{p}^{2}$	$RMSE_{p}$	$R_{p}^{2}$	$RMSE_{p}$	$R_{p}^{2}$	$RMSE_{p}$
NDVI	XGB	0.89	6.20	0.78	8.00	0.79	7.44	0.84	6.83	0.77	7.94	0.83	7.10
NDVI1	ET	0.65	8.12	0.50	11.00	0.65	10.21	0.73	9.04	0.65	10.2	0.63	10.50
NDVI3D	ET	0.83	6.72	0.73	8.90	0.78	8.14	0.83	7.20	0.77	8.22	0.79	8.20
VOG2	LGBM	0.83	6.95	0.71	9.30	0.77	8.36	0.82	7.42	0.09	16.4	0.82	8.50
MSR1	ET	0.86	7.09	0.68	9.70	0.73	8.95	0.78	8.16	0.73	8.98	0.73	8.80
SIPI	ET	0.55	10.10	0.39	13.55	0.48	12.45	0.56	11.52	0.36	13.8	0.40	13.05
MDATT1	ET	0.75	8.07	0.59	11.00	0.69	9.6	0.74	8.74	0.65	10.1	0.66	9.50
MCARI	LGBM	0.88	6.43	0.82	7.30	0.78	8.16	0.85	6.80	0.15	17.9	0.84	7.0
GRVI	ET	0.91	5.50	0.89	5.81	0.82	7.40	0.82	7.40	0.81	8.81	0.81	8.81
PRI	ET	0.18	18.20	0.31	17.19	0.14	15.77	0.19	15.54	0.01	17.1	0.20	15.05
VARI	LGBM	0.51	10.21	0.42	13.60	0.56	11.43	0.60	10.88	0.56	11.4	0.55	10.56
VREI	ET	0.87	6.15	0.77	8.24	0.78	8.16	0.83	7.12	0.77	8.25	0.78	8.18
SR	ET	0.85	7.75	0.82	7.39	0.82	7.64	0.84	6.88	0.80	7.66	0.82	7.80
RVSI	ET	0.49	12.50	0.72	14.72	0.52	12.02	0.54	11.75	0.32	17.9	0.55	8.72
BGI1	ET	0.10	20.3	0.39	20.41	0.20	17.45	0.20	17.45	0.17	18.6	0.15	19.23
Lic2	ET	0.65	10.21	0.55	11.00	0.65	10.17	0.73	9.04	0.65	10.1	0.66	10.23
PSRI	ET	0.59	8.50	0.48	12.4	0.60	10.99	0.62	10.69	0.30	14.4	0.58	13.56
NPCI	LGBM	0.28	19.5	0.13	18.37	0.14	16.01	0.14	16.01	0.01	17.1	0.30	14.50
CIgreen	ET	0.90	6.15	0.85	6.77	0.82	7.42	0.87	6.23	0.81	7.47	0.83	6.80

**Table 5 plants-13-00392-t005:** Evaluating color indices results.

VIs	AutoML			RF		PLSR		BPNN		SVM		R_Learner_	
	Model	$R_{p}^{2}$	$RMSE_{p}$	$R_{p}^{2}$	$RMSE_{p}$	$R_{p}^{2}$	$RMSE_{p}$	$R_{p}^{2}$	$RMSE_{p}$	$R_{p}^{2}$	$RMSE_{p}$	$R_{p}^{2}$	$RMSE_{p}$
rn	ET	0.65	8.0	0.41	13.0	0.20	18.0	0.55	9.0	0.31	12.81	0.25	13.81
gn	ET	0.79	5.0	0.43	12.5	0.23	15.2	0.63	9.5	0.33	11.47	0.29	12.29
bn	ET	0.59	10.0	0.39	12.9	0.19	13.6	0.51	11.5	0.27	15.9	0.23	16.4
GRRI	XGB	0.49	6.20	0.42	9.05	0.33	12.8	0.44	7.83	0.35	13.94	0.20	17.10
RBRI	LGBM	0.48	7.12	0.41	10.2	0.30	15.7	0.39	7.04	0.36	16.2	0.18	16.50
GBRI	RF	0.45	5.50	0.35	8.5	0.18	17.01	0.44	8.01	0.18	18.1	0.15	19.50
IKAW	XGB	0.85	2.05	0.35	11.85	0.32	13.29	0.49	10.56	0.13	19.08	0.19	17.88
NDI	ET	0.82	3.50	0.39	8.56	0.30	10.35	0.23	17.06	0.18	20.08	0.15	21.78
WI	ET	0.50	8.21	0.37	9.75	0.33	11.58	0.30	12.38	0.19	16.23	0.13	18.09
GLI	ET	0.81	7.54	0.34	10.65	0.32	12.23	0.29	13.20	0.23	15.05	0.18	20.89

## Data Availability

Data are contained within the article.

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
