# Peer review of "High-Throughput Analysis of Leaf Chlorophyll Content in Aquaponically Grown Lettuce Using Hyperspectral Reflectance and RGB Images"

_plants, 2024, doi:10.3390/plants13030392_

Round 1

Reviewer 1 Report (Previous Reviewer 1)

Comments and Suggestions for Authors

In this manuscript (plants-2848642), a model was developed using AutoML and other machine learning techniques to estimate the chlorophyll content in aquaponically grown lettuce, and spectral and colour indices were found to reliably estimate chlorophyll levels. The AutoML models outperformed the traditional models, with the combination of spectral and colour indices providing the most accurate predictions, demonstrating the potential for application in the control of the nutrient cycle in aquaponics.

The manuscript is suitable for publication. The topics of introduction, materials and methods, results, and discussion are appropriate. The images and tables are well formatted and set up. The text was written in clear and suitable language, but some frase need moderate changes in language.

Changes for “2. Material and Methods”

Please remove the side lines from the tables, especially from Table 4! The sample sizes have been added to the captions of the figures and tables. What was the applied confidence interval? Make this clear in the caption.

Comments on the Quality of English Language

Moderate editing of English language required

Author Response

In this manuscript (plants-2848642), a model was developed using AutoML and other machine learning techniques to estimate the chlorophyll content in aquaponically grown lettuce, and spectral and colour indices were found to reliably estimate chlorophyll levels. The AutoML models outperformed the traditional models, with the combination of spectral and colour indices providing the most accurate predictions, demonstrating the potential for application in the control of the nutrient cycle in aquaponics.

The manuscript is suitable for publication. The topics of introduction, materials and methods, results, and discussion are appropriate. The images and tables are well formatted and set up. The text was written in clear and suitable language, but some frase need moderate changes in language.

Authors: We appreciate your valuable time to review the manuscript. Thank you for recognizing the contribution of our work. We appreciate all your valuable comments and suggestions that enriched the article. Indeed, we have corrected many linguistic errors throughout the manuscript.

  1. Changes for “2. Material and Methods”

Authors: Thank you for this comment, but we followed the journal’s template, which states that it is written “Materials and Methods.”

  1. Please remove the sidelines from the tables, especially from Table 4! The sample sizes have been added to the captions of the figures and tables. What was the applied confidence interval? Make this clear in the caption.

Authors: The sidelines were removed from the tables and a confidence interval was added to the caption of the figures.

Reviewer 2 Report (Previous Reviewer 3)

Comments and Suggestions for Authors

The authors report a study using machine learning to monitor the growth processes of lettuce crops. Specifically, machine learning systems were used to monitor chlorophyll and other accessory pigments in aquaponic lettuce crops. 

The authors submitted the manuscript with many revisions. In my opinion, the manuscript has improved from the previous version. However, some additions are needed:

1) Figure 1 is unclear. Is there a microbiological control on the recovery of water from the aquarium? Has the bacterial load in the lettuce been measured?

2) It is unclear to me whether fish in the aquarium are involved in the experiment. If your experiment involves animals, it is essential to include an official permit from the university institution and the Ministry of Health for the use of animal species. 

3) The authors should include a paragraph where they explain the state of fish breeding. Specifically, species used, environmental conditions of growth, duration of the entire life cycle etc...

4) Although the topic is interesting for "precision agriculture", I wonder about the feasibility for other greenhouse or field crops. Can your machine learning system be applied to wheat, corn, barley crops in the field? A comment on these questions should be included in the conclusion. 

5) Finally, I suggest you comment on this recent manuscript on photosynthetic processes: Makhtoum, S. et al. Genomics and physiology of chlorophyll fluorescence parameters in Hordeum vulgare L. under drought and salt stress. Plants 2023, 12, 3515. https://doi.org/10.3390/plants12193515

Comments on the Quality of English Language

 Minor editing of English language required

Author Response

The authors report a study using machine learning to monitor the growth processes of lettuce crops. Specifically, machine learning systems were used to monitor chlorophyll and other accessory pigments in aquaponic lettuce crops.

The authors submitted the manuscript with many revisions. In my opinion, the manuscript has improved from the previous version. However, some additions are needed:

Authors: Thank you indeed for the effort and time devoted in reading and commenting on the manuscript and for your constructive comments.

  1. Figure 1 is unclear. Is there a microbiological control on the recovery of water from the aquarium? Has the bacterial load in the lettuce been measured?

Authors: Thank you for this comment, Indeed, after the water leaves the fish tank, it goes through multiple purification stages, including the biological filter with its components that stimulate the growth of nitrifying bacteria, which converts ammonia into nitrite and then into nitrate. The plant absorbs nitrates and other nutrients to return the water clean and pure again to the fish tank. The microbial load of plants growing in aquaponic systems is usually not measured. In addition, the microbiological analyses were not one of the goals of this study but is it a noteworthy point to consider in our future research endeavors.

  1. It is unclear to me whether fish in the aquarium are involved in the experiment. If your experiment involves animals, it is essential to include an official permit from the university institution and the Ministry of Health for the use of animal species.

Authors: Fish are an essential component of the system, all experiments of this study were carried out according to the regulations of the university.

  1. The authors should include a paragraph where they explain the state of fish breeding. Specifically, species used, environmental conditions of growth, duration of the entire life cycle etc...

Authors: Thank you for this comment, This is included in the manuscript, lines 163-167.

  1. Although the topic is interesting for "precision agriculture", I wonder about the feasibility for other greenhouse or field crops. Can your machine learning system be applied to wheat, corn, barley crops in the field? A comment on these questions should be included in the conclusion.

Authors: We appreciate this comment, this is included in the conclusion lines 652-654.

  1. Finally, I suggest you comment on this recent manuscript on photosynthetic processes: Makhtoum, S. et al. Genomics and physiology of chlorophyll fluorescence parameters in Hordeum vulgare L. under drought and salt stress. Plants 2023, 12, 3515. https://doi.org/10.3390/plants12193515.

Authors: Since this is a recent article, it has been cited that demonstrates the importance of chlorophyll as an indicator of salinity and drought stresses.

Round 2

Reviewer 2 Report (Previous Reviewer 3)

Comments and Suggestions for Authors

The authors greatly improved the manuscript. In my opinion it can be published.

This manuscript is a resubmission of an earlier submission. The following is a list of the peer review reports and author responses from that submission.

Round 1

Reviewer 1 Report

Comments and Suggestions for Authors

This manuscript (plants-2791255), a model was developed using AutoML and other machine learning techniques to estimate the chlorophyll content in aquaponically grown lettuce, and spectral and colour indices were found to reliably estimate chlorophyll levels. The AutoML models outperformed the traditional models, with the combination of spectral and colour indices providing the most accurate predictions, demonstrating the potential for application in the control of the nutrient cycle in aquaponics.

The manuscript is suitable for publication. The topics of introduction, materials and methods, results, and discussion are appropriate. The images and tables are well formatted and set up. The text was written in clear and suitable language.

Minor corrections and suggestions for the authors.

Keywords in alphabetical order;

Which lettuce cultivars were used?

What was the standard, and what were the procedures for calibrating the FieldSpec? Which probe was used? What was the diameter?

What was the rationale for using a 70:30 ratio instead of 75:25 or 60:40? How many lettuce samples were analysed? And how many spectra were collected per leaf and recorded on the equipment?

Figures containing colored elements should be referenced or accompanied by a legend for proper explanation. What was the actual sample size? Please describe (n=x);

Consider updating some references earlier than 2015 with more recent ones.

Comments on the Quality of English Language

Minor changes in the English language are needed.

Reviewer 2 Report

Comments and Suggestions for Authors

This research explored the viability of creating an automated machine learning (AutoML) pipeline for estimating chlorophyll content of aquaponically cultivated lettuce, utilizing spectral and color vegetation indices. Although authors got satisfactory results, this manuscript appear very poor in writing. This manuscript did not have discussion and it was necessary to separate the discussion from the results. And in added discussion, authors need to explain the advantage of AutoML and why combined spectral and color vegetation indies can improve the accuracy. Because I think the discussion about this wrote in results was not enough. This manuscript has lots of inappropriate expression and grammatical error. Maybe the author could get help from native English speaker to revise them.

Line 23: “state-of-the-art” is not appropriate. Revise it.

Line 79: “An image dataset is obtained, then segmented, and features associated with chlorophyll content are extracted” The statement of the sentence is inconsistent with the context. Please revise it.

line 151: It is recommended to draw a schematic diagram of the aquaponic system and label the parameters.

Line226,346: vegetation indices. The abbreviation must be the same. Currently, it is Vis or VI.

Line 361-362: I did not understand why authors compared the results with NDVI. Because NDVI by using your AutoML also could bring better accuracy of 0.89. And in this part of discussion, authors should give us a detailed and specific explanation about the advantage in estimation of chlorophyll content not nitrogen content.

Line 364: How do you calculate R2p RMSEp? Please supplement the calculation formula.

Line 364:  Table4. The expression of the table is confused, and the position of the model column of AutoML is unreasonable. For example, GRVI will be considered as R2p=0.89 when ET and RF are combined, and R2p=0.82 when ET and PLSR are combined.

Line 371: Figure 4 should be revised to figure 4 (d, e and f). Please check other similar mistakes.

Line 387. What are R2c and RMSEc in Figure4, please explain and give the calculation formula.

Line387. Figure4, line422 Figure5, line442 Figure6, line499 Figure7, The color and shape of the scatter diagram are not good, please modify it.

Line 499. vegetation indices in the figure are "NDI" and "gn" in larger font.

Line 520-521: “Notably, color images focus more on phenotypic traits, while spectral data look at plant physiology”. Please explain why?

Line 529-571: This paragraph should appear in discussion.

Line 578: What is the meaning of “the top”? The best??

Comments on the Quality of English Language

This manuscript has lots of inappropriate expression and grammatical error. Maybe the author could get help from native English speaker to revise them.

Reviewer 3 Report

Comments and Suggestions for Authors

The authors report a simulation study of chlorophyll levels in hydroponic lettuce crops. 

The manuscript could have utility in agronomy and food science. However, in my opinion, fundamental aspects are missing that suggest me to reject the amnuscript. 

Following are some points:

Why did the authors evaluate only the chlorophyll parameter?

It is not clear with how many samples the system was trained. This aspect should be clarified.

Can the predictive model be applied to other crops as well?

The authors should have "trained" the model with other plant species as well.